# Spatially confined protein assembly in hierarchical mesoporous metal-organic framework

Xiaoliang Wang [1,2], Lilin He [3] ✉, Jacob Sumner[3], Shuo Qian [3,4], Qiu Zhang[3], Hugh O'Neill [3], Yimin Mao[5,6], Chengxia Chen[1], Abdullah M. Al-Enizi [7], Ayman Nafady[7] & Shengqian Ma [1] ✉

Immobilization of biomolecules into porous materials could lead to significantly enhanced performance in terms of stability towards harsh reaction conditions and easier separation for their reuse. Metal-Organic Frameworks (MOFs), offering unique structural features, have emerged as a promising platform for immobilizing large biomolecules. Although many indirect methods have been used to investigate the immobilized biomolecules for diverse applications, understanding their spatial arrangement in the pores of MOFs is still preliminary due to the difficulties in directly monitoring their conformations. To gain insights into the spatial arrangement of biomolecules within the nanopores. We used in situ small-angle neutron scattering (SANS) to probe deuterated green fluorescent protein (d-GFP) entrapped in a mesoporous MOF. Our work revealed that GFP molecules are spatially arranged in adjacent nanosized cavities of MOF-919 to form "assembly" through adsorbate-adsorbate interactions across pore apertures. Our findings, therefore, lay a crucial foundation for the identification of proteins structural basics under confinement environment of MOFs.

The emergence of metal-organic frameworks (MOFs), which are periodically constructed by the assembly of a great variety of metal ion/metal ion cluster nodes and multitopic organic ligands, rapidly attracts great attention as solid matrices for biomolecules encapsulation[1-5]. Compared with classic porous materials, the large pore size and adjustable pore structure together with high surface area synergically endow MOFs with exceptionally high loading capacity for biomecules[6-9]. Despite enormous efforts that have been devoted to the exploration of stability, and/or activity of the biomolecules immobilized into MOFs, the confinement effects of porous supports on the conformations of protein molecules and their spatial arrangement remain elusive[10,11]. There are long-existing challenges for obtaining structural information of the biomolecules entrapped in MOFs[12-18]. Usually, the measurement of protein structures is conducted using conventional spectroscopic analysis, such as solid-state UV–visible spectrophotometry, Raman, and FTIR[19-24]. Surface analytical techniques are also useful to monitor immobilization processes[23]. However, these phenomenal techniques generally give indirect evidence to probe the confinement-induced conformational changes or dynamic constraints due to the difficulty of spectral assessment and complexity of protein-protein or protein-matrix interactions, and they are difficult to provide the direct depiction of their arrangement or overall structure under confinement environment.

[1]Department of Chemistry, University of North Texas, Denton, TX 76201, USA. [2]Department of Chemistry, University of South Florida, Tampa, FL 33620, USA. [3]Neutron Scattering Division, Oak Ridge National Laboratory, Oak Ridge, TN 37831, USA. [4]The Second Target Project of SNS, Oak Ridge National Laboratory, Oak Ridge, TN 37831, USA. [5]NIST Center for Neutron Research, National Institute of Standards and Technology, Gaithersburg, MD 20899, USA. [6]Department of Materials Science and Engineering, University of Maryland, College Park, MD 202742, USA. [7]Department of Chemistry, College of Science, King Saud University, Riyadh 11451, Saudi Arabia. ✉e-mail: hel3@ornl.gov; Shengqian.Ma@unt.edu

Small-angle neutron scattering (SANS), capable of detecting length scales ranging from ~1 to 200 nm, is uniquely suited for probing the structures of biomolecules in matrix mesoporous materials[25–31]. More importantly, it can minimize the scattering contribution from the solid matrices by tuning scattering contrast of solvent with the variation of deuterated and hydrogenated solvents ([1]H, neutron coherent scattering length $b_c = -3.742$ fm; deuterium, D or [2]H, neutron coherent scattering length $b_c = 6.675$ fm). It is targeted to selectively match the neutron scattering length density (NSLD) of the solid matrix and allow the extraction of structural parameters to determine the overall structure of biomacromolecules from a multi-component system[32,33]. The scattering intensity is proportional to the square of the Fourier Transform of the scattering object's NSLD distribution, averaged over all orientations. The azimuthally averaged 1D curve plotted against scattering angle (or q-vector) carries the information of the molecular weights, dimensions, and low-resolution shapes. Further structural details and interactions can be obtained via model fitting and/or ab initio 3D shape reconstruction.

Employing the SANS technique, we were able to for the first time determine the protein spatial arrangement within the MOF nanopores as illustrated in the present work. Green fluorescent protein (denoted as GFP) was selected as the model protein. Considering the dimension of GFP (a cylinder with a length of 4.2 nm and a diameter of 2.4 nm, Supplementary Fig. 1), in this protocol, a large but water-stable mesoporous MOF with zeolite MTN topology MOF-919 was chosen as host matrix, which was constructed by ditopic ligand of 1H-pyrazole-4-carboxylic acid ($H_2PyC$) and vertices of two types of metal-containing second building units (SBUs), Al-SBU and Cu-SBU, respectively[34]. MOF-919 was reported containing one microcage of 1.8 nm in diameter (Fig. 1a), and two mesocages of 4.9 nm and 6.0 nm, which two types of mesocages are fused through pentagonal aperture of 2.0 nm, and 2.4 nm hexagonal openings for two large *liu* cages, respectively (Fig. 1b–d and Supplementary Figs. 2 and 3). To better determine the backbone arrangement of accommodated GFP, free GFP in solution, pristine MOF-919 and MOF-919 loaded with GFP (denoted as

GFP@MOF-919) at matching point of MOF, were measured by SANS, separately (See schematic in Supplementary Fig. 4). The preparation of all samples including pristine MOF-919 and GFP@MOF-919, are provided in 'Methods'.

## Results

The SANS profile of the empty MOF-919 was collected at dry state to reveal the pore morphologies. The SANS instrument configurations were selected to cover a q-range of $0.0015 < q < 0.5$ Å$^{-1}$. The scattering of the dry material exhibits three independent contributions owing to its hierarchical structures (Fig. 1e, the red line). The regime I (Fig. 1e, the blue shade), in the low q region of 0.003–0.02 Å$^{-1}$, follows a surface fractal, indicated by the power law decay with an exponent of −3.4. The surface fractal consists of scattering units of different sizes following a power-law distribution. The regime II (Fig. 1e, the orange shade), in the q range of 0.02–0.06 Å$^{-1}$, arises from the material inhomogeneity[35,36]. The regime III (Fig. 1e, the gray shade), in the high q range of 0.06–0.25 Å$^{-1}$, reveals the ordered pore arrangement. The best fitting of the full curve using a summed model of a power law for the surface scattering, correlation length model for material's inhomogeneity, Gaussian functions for Bragg peaks, and the incoherent scattering background is shown as the solid line in Fig. 1e.

$$I(q) = \frac{A}{q^n} + \frac{B}{1+(q\xi)^m} + \sum_{i=1}^{3} C_i e^{-\frac{(q-q_{0i})^2}{2D_i^2}} + I_{bgd} \quad (1)$$

where $A$, $B$, $C$ are the prefactors, $n$ is the exponent of the power law decay, $\xi$ is the correlation length describing the average size of the material inhomogeneity. The fitting reveals that the average size of this structural feature is ~265 Å by Eq. (1), and it is likely to be associated with the polydispersed macropores or large-scale inhomogeneity in the material. In the Gaussian function, $q_{0i}$ is the center of $i$th peak and $D_i$ is related to the half-width-half-maximum (HWHM) of the peak. $I_{bgd}$ represents the incoherent background. The fitting parameters are listed in the table (Supplementary Table 1).

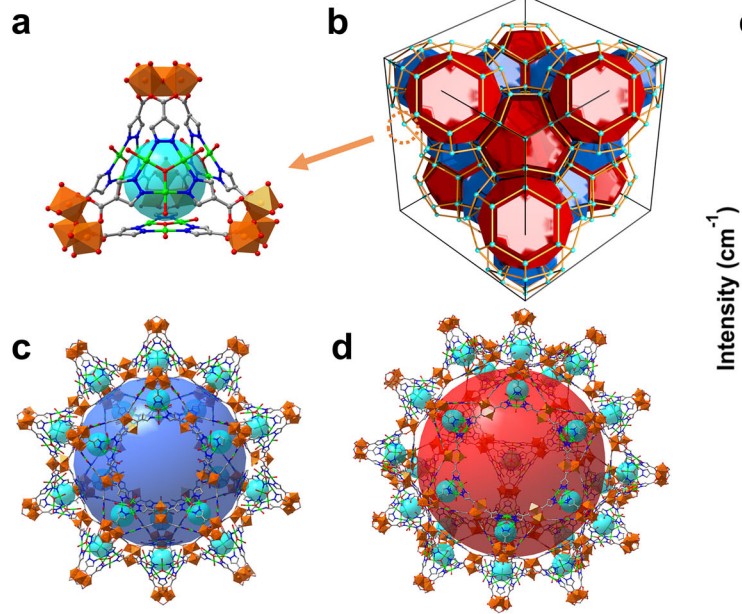

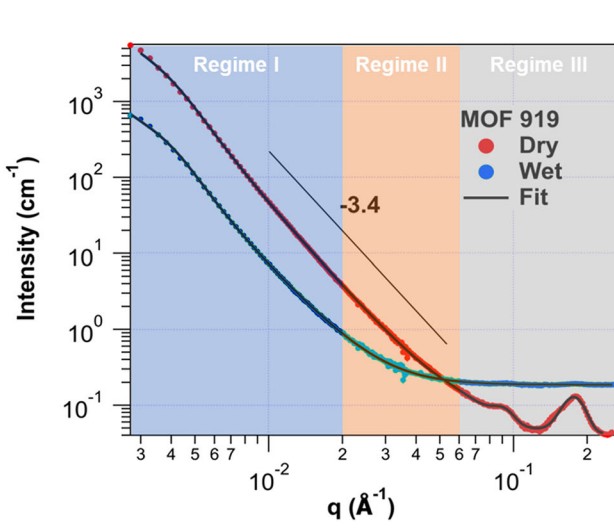

**Fig. 1 | Structure and SANS profiles of MOF-919. a** Microporous cage (cyan sphere). **b** Topology with unit cell (cyan: microcage, blue: small mesocage, red: large mesocage). **c** Small *yys* mesocage (large blue sphere). **d** Large *liu* mesocage (large red sphere). **e** SANS profiles of dry MOF-919 and MOF-919 in the contrast matching point of 50% $D_2O$ and 50% $H_2O$ mixture (wet). The solid lines corresponding to the best fitting using the models mentioned in the text. The q range of three regimes: Regime I (blue shade, 0.003–0.02 Å$^{-1}$), Regime II (orange shade, 0.02–0.06 Å$^{-1}$), Regime III (gray shade, 0.06–0.25 Å$^{-1}$). Color scheme (C: gray; N: blue; O: red; Cu: green; Al: brown).

## Contrast variation experiment

Neutron contrast depends on the isotopic composition of scattering objects and surrounding media. Thus, contrast variation is a uniquely powerful feature of neutron scattering, which is inaccessible to other conventional techniques[26,32]. NSLD of MOF-919 was calculated as $3.15 \times 10^{-6} \text{Å}^{-2}$, based on the skeletal density of $1.783 \, \text{gcm}^{-3}$ (porosity, 80.7%). In order to obtain the exact contrast matching point of matrix MOF, mixtures with different volume ratios of $H_2O$ and $D_2O$, were used to saturate the composites. The contrast matching ratio of 50% $D_2O$ by volume, was subsequently determined, which is in accordance with the calculated matching point (54% $D_2O$). At contrast matching point, the background scattering signal of MOF-919 was matched out to specifically highlight the protein scattering (the blue line in Fig. 1e). Compared with the SANS profile of dry MOF-919 (the red line in Fig. 1e), the high q peaks of unloaded MOF in water vanished, thereby indicating that the majority of ordered and highly monodispersed pores are accessible to water. The scattering intensity in the low q region dropped due to the penetration of solvent into the large-scale disordered pores. The residual scattering arose from inaccessible mesopores and large-scale density inhomogeneity that can be described by the surface fractal dimension. An appreciable increase of background can be attributed to the incoherent scattering of $H_2O$ in the mixture solvent. Thus, the contrast matching method helps to enhance the scattering signal of protein component. The SANS profile of MOF-919 at contrast matching point can be modeled by Eq. (1) without the Gaussian terms for the Bragg peaks (Fig. 1e and Supplementary Table 2). Interestingly, the correlation length minimally changed upon the water penetration, which further confirms that the low q diffuse scattering is attributed to the overall material inhomogeneity at large scale less relevant to protein encapsulation.

The ubiquitous existence of hydrogen in soft matter and biomolecules enables the utilization of isotope deuterium to enhance the contrast, by replacing H with D during the protein expression without altering their structures and chemical properties[37–39]. The reported contrast matching value of hydrogenated GFP (h-GFP) is 42% $D_2O$, and it is close to the calculated SLD of matrix MOF-919 (~54% $D_2O$) that is also in good agreement with the measured contrast matching point (~50% $D_2O$, Supplementary Fig. 5 and Supplementary Table 2). Considering the low scattering contrast between h-GFP and backbone of

MOF-919, deuterated GFP (d-GFP) with a much higher contrast matching point than h-GFP, was used here. Siefker et al. previously observed spatial orders of fluid-like structure factor at maximum protein concentration in SBA-15[40]. Thus, to investigate the effects of protein concentration on its distribution inside hierarchical MOF-919, a variety of d-GFP@MOF-919 composites were prepared via loading in decreasing concentration of d-GFP in 20 mM Tris-buffer (pH 7.5, 50 mM NaCl) at 18.6, 9.6, 6.2, and 3.8 mg/mL (**C1**−**C4** hereafter), respectively, under ambient conditions. The loading efficiency of GFP reached 81.7% (calculated to be 12.3 μmol/g) after 12 h for the representative group (Supplementary Fig. 6). After washing d-GFP@MOF-919 composites (**C1**−**C4**) to remove surface attached protein, the composite exhibits strong fluorescence under fluorescent microscopy, indicating the successful loading of GFP in MOF-919. It also shows decreasing intensity in fluorescence (**C1**−**C4**) under the same conditions as the initial GFP concentration was decreased (Supplementary Fig. 7). It is observed in FT-IR spectra that the emergence of shoulder peak at 1630 $\text{cm}^{-1}$ of GFP@MOGF-919 composites, which can be rationalized by the existence of GFP associated with amide I band (Supplementary Fig. 8). The powder X-ray diffraction (PXRD) pattern of MOF-919 before and after the inclusion of GFP confirmed the bulk crystallinity was retained after protein loading (Supplementary Fig. 9). Scanning electron microscopy (SEM) of GFP@MOF-919 composites show no apparent changes on particle morphology after the loading of GFP under different concentration (Supplementary Fig. 10). Note that all concentrations hereafter indicate the original d-GFP concentration where the d-GFP@MOF-919s were prepared.

Figure 2a shows the SANS profiles of empty MOF-919 and d-GFP@MOF-919s at the contrast matching condition of MOF-919. Notably, SANS confirmed the successful loading of d-GFP evidenced by the significant scattering decay of high q range at dry state (Supplementary Fig. 11), due to the raised scattering contrast between the matrix MOF and the protein-filled pore space. Upon the loading of d-GFP into the mesopores of MOF-919, the scattering intensity showed dramatic increase in the whole q range compared with that of empty MOF in 50% $D_2O$ (Fig. 2a). Also, the SANS signal increased with increasing loading protein concentration while a broad shoulder in the mid q region near ~0.04 $\text{Å}^{-1}$ developed at the wet state (indicated by orange arrow in Fig. 2a), which is the signature of the hosted d-GFP (Supplementary Fig. 12), and the SANS signal from accommodated

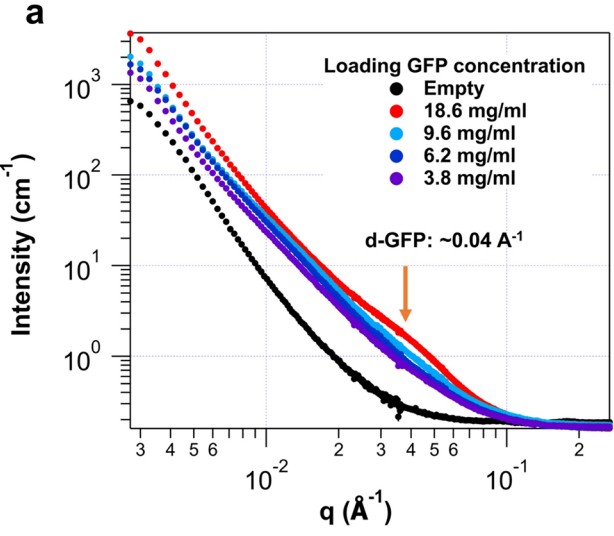
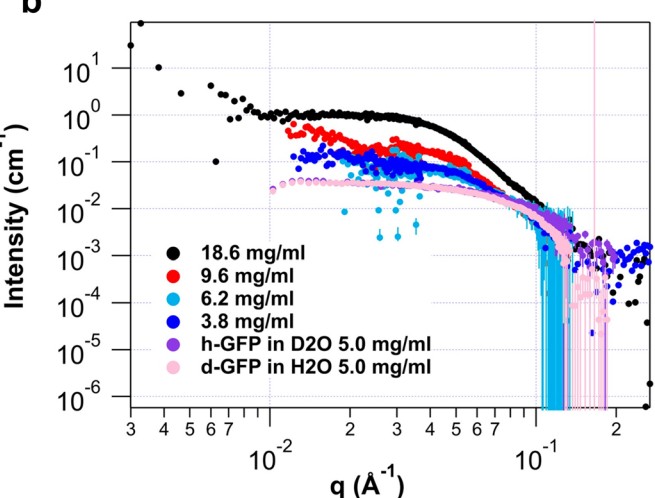

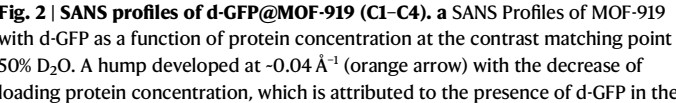

**Fig. 2 | SANS profiles of d-GFP@MOF-919 (C1−C4). a** SANS Profiles of MOF-919 with d-GFP as a function of protein concentration at the contrast matching point 50% $D_2O$. A hump developed at ~0.04 $\text{Å}^{-1}$ (orange arrow) with the decrease of loading protein concentration, which is attributed to the presence of d-GFP in the nanopores. (Initial loading concentration of d-GFP. **C1**: 18.6 mg/mL, **C2**: 9.6 mg/mL, **C3**: 6.2 mg/mL, and **C4**: 3.8 mg/mL). **b** Scattering of GFP confined in MOF-919 after the subtraction of MOF scattering. The scattering of h-GFP in $D_2O$ and d-GFP in $H_2O$ are plotted for the comparison.

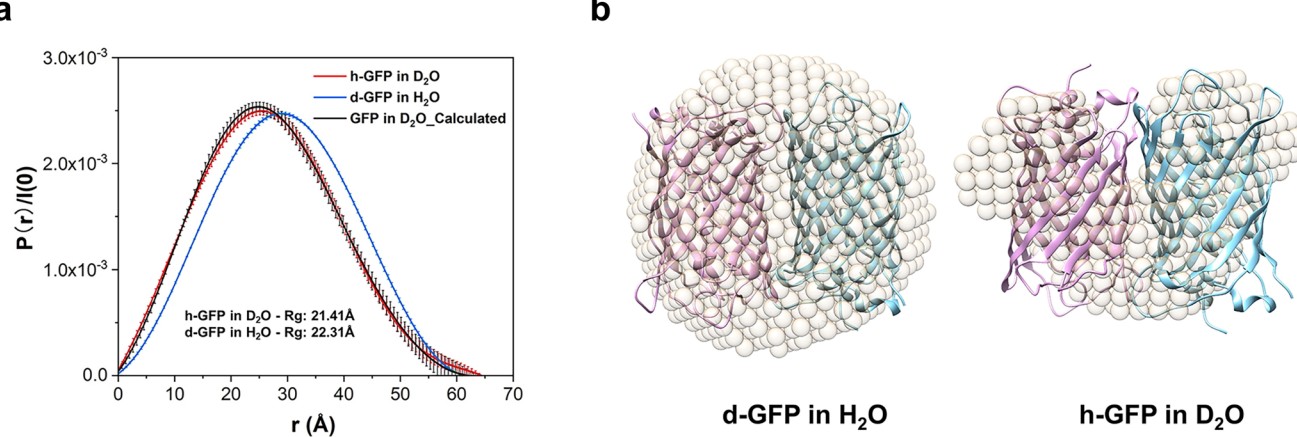

**Fig. 3 | Reconstruction of free d-GFP and h-GFP. a** Pair distance distribution function (PDF or P(r)) curves for free d-GFP in $H_2O$ and h-GFP in $D_2O$ and calculated P(r) with GFP (PDB:1GFL) for comparison. **b** DAMMIF reconstruction of d-GFP in $H_2O$ and h-GFP in $D_2O$. The reconstruction by scattering profiles is represented by light gold beads and the PDB structure of GFP is overlaid for validation.

d-GFP was at least one order of magnitude lower than the matrix MOF. Thus, at contrast matching point, the residual MOF scattering signal was subtracted to provide the scattering signal only from the accommodated d-GFP.

The scattering intensity of d-GFP hardly changes when the loading concentration of protein increases from 3.8 mg/mL to 6.2 mg/mL. The intensities of the samples with the loading concentrations of 9.6 mg/mL and 18.6 mg/mL are significantly stronger (Fig. 2b). The Guinier fit also indicates that the proteins are in large "cluster" with higher loading concentrations[41]. To compare the conformation of proteins confined in mesopores and that of free GFP in solution, free h-GFP in $D_2O$ and d-GFP in $H_2O$ at 5 mg/mL were also analyzed with SANS. The radii of gyration (Rg) for both proteins are 21.41 Å and 22.31 Å, respectively. This is larger than the Rg of d-GFP as monomer (using PDB structure) in $H_2O$ and h-GFP in $D_2O$, 16.82 Å and 18.90 Å from calculated scattering curve by CRYSON, respectively (see Supplementary Table 4). The pair distance distribution function P(r) reveals that the most probable pair distance of the d-GFP in $H_2O$ shifts toward larger ones in comparison with that of h-GFP in $D_2O$ although the maximum real-space dimension $D_{max}$ nearly coincides (Fig. 3a). This has been ascribed to the contrast difference between the protein and the hydration shell[37]. The symmetrical bell-shaped peak of free d-/h-GFP in solution is indicative of a spherical compact particle. Ab initio reconstruction using DAMMIF model, including FAST and SLOW MODE indicated that the shape and dimensions of free d-GFP and h-GFP molecules in solution are similar (Fig. 3b and Supplementary Figs. 13 and 14), in a noncovalently loose dimer configuration known previously[42,43]. Note this free GFP dimer is different from the protein cluster or unit formed with constraint of MOFs discussed later.

The Rg value of the confined protein depends on the concentration (Fig. 4a and Table 1). **C3** and **C4** show similar Rg value (23.01 and 24.63 Å, respectively), both of which are slightly larger than the Rg of free d-GFP in solution (22.31 Å). **C1** and **C2** exhibit significantly larger Rg (39.70 and 30.69 Å, respectively), which reflects the cluster due to the close arrangement among proteins at higher concentrations. This observation is also shown by the P(r) analysis that evolves from the bell shape to more complex distribution with the increase of the concentration (Fig. 4a). The presence of the Guinier plateau in the scattering profiles indicates that the "assembly" formed by the confined proteins is not hierarchical and free of network structure. The missing hump in the same q range of the SANS profile of the empty dry MOF (Fig. 1e the red line) also indicates that the presence of partial-ordered defect pores in the same length scale is unlikely. Presumably, the protein-protein interactions induced by confinement exists among

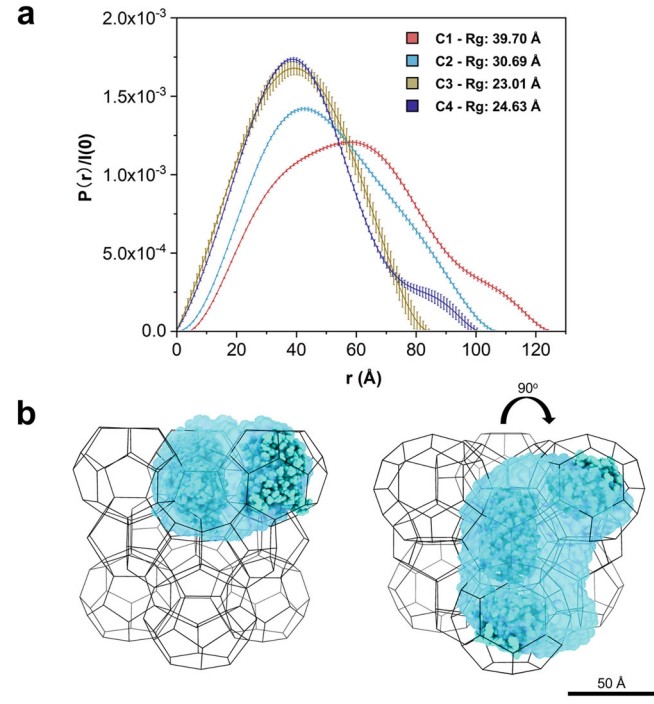

**Fig. 4 | 3D reconstruction of d-GFP@MOF-919. a** Experimental P(r) curves for d-GFP loaded in the MOF-919 (**C1–C4**). **b** Ab initio structure reconstruction of **C2** in structure of MOF-919. The blue-shaded region is reconstructed using scattering profile of **C2**, and the GFP atomic structure is overlaid by blue balls for comparison.

**Table 1 | Parameters of P(r) curves using d-GFP@MOF-919 (C1–C4)**

|        | Peak a (Å) | Peak b (Å) | Peak c (Å) | Rg (Å) | $D_{max}$ (Å) |
|--------|------------|------------|------------|--------|---------------|
| **C1** | 32.90      | 58.61      | 100.77     | 39.70  | 126           |
| **C2** | 42.58      | –          | –          | 30.69  | 108           |
| **C3** | 39.20      | –          | –          | 23.01  | 85            |
| **C4** | 38.64      | 85.78      | –          | 24.63  | 102           |

accommodated proteins in adjacent cages through apertures to form such "assembly". This behavior was also observed at the dry condition, exhibiting similar feature in the scattering patterns (Supplementary Fig. 11).

## 3D reconstruction of d-GFP in MOF-919

Going further in visualization of protein arrangement confined in the hierarchical structure of MOF-919, all d-GFP@MOF-919s were reconstructed with ab initio method implemented in DAMMIF[44]. Loaded inside MOF cavities, it is intriguing that the protein at **C1**–**C4** concentrations is not denatured but maintaining its native conformation, supported by Kratky plots (Supplementary Fig. 15). This suggests the conformation of immobilized proteins in MOF can be well-maintained. Once the ab initio results of the envelope were obtained, the PDB structure of GFP is filled into the volume with the constraint of MOF cage structure and layout to further mimic the shape and dimension of d-GFP in MOF-919 at contrast matching point. Then the simulated GFP 'multimer' configuration is used to generate calculated scattering with CRYSON. The P(r) curves were obtained using the calculated scattering. These multimer were created and assumed the proteins in the confined nanopores of MOFs as monomer, dimer (side-by-side and perpendicular), trimer and tetramer (Supplementary Table 4 and Supplementary Fig. 16). The P(r) parameters in Table 1 are calculated from Fig. 4a and the multiple peaks at larger r value represent the existence of different oligomers. The multimer in Table 1 is defined as the spatially confined protein assembly, consisting of multiple proteins isolated evenly in mesopores of MOF-919. It is conceptually different from a protein complex oligomer which formed by the interaction of several individual proteins via non-covalent bond, for example the dimer of free GFP we discussed previously.

Based on the reconstructions and the maximum dimension in P(r) analysis, the size of GFP particles in **C1** and **C2** samples is much larger than a GFP monomer, likely an oligomer or a cluster GFP encaged (or constrained) by the MOF lattice. For example, **C1** shows three peaks at 32.90 Å, 58.61 Å, and 100.77 Å and **C2** has a peak at 42.58 Å but with a wide shoulder around 80 Å (see Table 1). The size and shape of protein assembly in **C2** are comparable to "trimers" or a mix of "trimers" and "dimers", shown as the blue-shaded region overlaid with GFP atomic structure and matrix MOF-919 in Fig. 4b, in which GFP molecules are arranged in three adjacent mesopores (Supplementary Fig. 19). However, **C1** is more likely to form larger group like "tetramers" among MOF cavities and tends to cover four adjacent cages via the reconstruction (Supplementary Fig. 18). **C3** and **C4** exhibit more similar arrangement to each other, and the reconstruction reveals a GFP assembly in an interesting configuration, resembling a GFP monomer perpendicular to another GFP (Supplementary Figs. 20 and 21), due to the MOF constraint in geometry. It is noted that, with the increasing loading of GFP in MOF structure, all arrangements have a peak (or hump) representing the GFP assembly around 32–42 Å (Peak a in Table 1 and Supplementary Figs. 18–21), while the most probable pair distance of the free d-GFP calculated in solution is peaked at around 25–30 Å as shown in Fig. 3. This indicates that the protein assembly is different within solution due to the confined environment of MOF, which is also supported by the divergent Porod volume shown in Supplementary Table 5. The Porod volume for d-/h-GFP monomers are smallest ($36.5 \times 10^3$ and $30.2 \times 10^3$ Å$^3$) and the experimental free d-GFP in H$_2$O ($61.1 \times 10^3$ Å$^3$ side-by-side dimer) have a larger volume and nearly twice of the volume than d-GFP (PDB structure, monomer) in H$_2$O. However, **C1** and **C2** exhibits much larger volumes than a trimer and tetramer and it is believed that it not easy to accurately identify the assembly level simply based on Porod volumes especially for the larger protein assemblies (>dimer) in MOFs while considering the gap volume among proteins. The likelihood of larger GFP assemblies being captured increases with concentration, since **C1** and **C2** samples have higher loading concentrations of GFP and therefore have larger length scales captured in the P(r) distribution above. In addition, it is very intriguing that the best-fitted arrangement of entrapped GFP is consistent with the distribution of large mesocage of MOF-919 (Supplementary Fig. 3), exhibiting the likely preferred accommodation in larger mesopores of MOF-919. It was observed that BET surface area dropped from 2168 to 1639 m$^2$/g after GFP

encapsulation, and pore size distribution analysis (PSD) indicated pore volumes of both nanoscale pores decreased due to the presence of entrapped GFP, and micropore remains unchanged (Supplementary Fig. 17). However, limited by nitrogen adsorption method, it is unlikely to distinguish the preferred entrance of protein in single group of mesopores only based on the pore dimension analysis in this work, especially for the pores with similar dimension. Although many reports indicate that GFP is prone to forming non-covalent dimers in solution, as well as shown by our free GFP data, the multimer or protein assembly in this work are conceptually different as a hierarchical organization imposed by MOF structure[45–47]. To illustrate, the multimers represent the confined protein clusters in MOF-919 (Supplementary Table 4), and the dimer is two d-GFP assembly formed next to each other in MOFs cage therefore it shows larger Rg, 30.73 Å (side-by-side) and 31.06 Å perpendicular with each other. The trimer/tetramer formed by triple/quadruple GFP exhibits Rg values of 40.75 Å and 46.04 Å, respectively. Combined, the dimensions of proteins in MOF are proved much larger than noncovalently bounded GFP assembly in solution. However, the geometric effect of MOF likely confined the accommodated GFP in each single cavity resulting larger dimension than a monomer, as shown by the reconstructed 3D structures of **C3** and **C4** (Supplementary Figs. 20 and 21). The subtle differences were observed while the initial loading concentrations of protein are low. Comparing with the calculated of multimers in MOFs, the smaller experimental Rg (**C1**–**C4**) indicates the smaller distance among proteins. Thus, it is believed that the existence of protein-protein interactions and confinement effects synergistically resulted in the closer protein arrangement. GFP protein and its derivatives are widely used as noninvasive markers in many biomedical and bioengineering research. It is relatively stable with the beta barrel structure, yet we have found MOF is able to encapsulate the protein and form stable assemblies within the nanospace. The availability of deuterated GFP allowed this study to be done and the new knowledge about its distribution in the MOF can be applied to different types of proteins and we would expect similar behaviors from other similar proteins.

## Discussion

To summarize, it was demonstrated that SANS can directly visualize encapsulated protein in the nanopores of MOF. Contrast matching method along with d-GFP provided a strong scattering signal, and made it possible to distinguish the protein component from hydrogenated MOF at contrast matching point. For all cases, the scattering profiles at high q range probe the protein assembly inside pores. The existence of abundant large openings surrounding cages and bridging two mesopores were considered to allow the formation of protein "assembly", particularly through the cage apertures of MOF-919. To our knowledge, this is the first work to unveil the protein spatial arrangement in MOFs via SANS, and the utilization of fully deuterated protein offered higher neutron scattering that overcame the limitation of low scattering contrast between the MOF and protiated protein. The highly ordered structure features of matrix MOF are all well-presented by SANS, and it enables the ease in characterization and modeling protein overall structures. The unique feature of contrast matching allows the effective extraction of structure information of confined proteins, in situ selectively from the host matrix MOF-919. The clarification of protein behaviors or arrangements in confinement environment will assist the understanding of protein performance under various conditions. More importantly, it is a crucial attempt to develop novel and facile approach to probe the behaviors of proteins and other biomolecules in the nanospace of MOFs, which will be of significance for applications in biocatalysis, biomedicine, and beyond.

## Methods
### Chemicals and characterization instruments
N,N-dimethylformamide (DMF), Copper nitrate trihydrate (Cu(NO$_3$)$_2$·3H$_2$O), trifluoroacetic acid (TFA), tris(hydroxymethyl)

aminomethane (Tris) were purchased from Fisher Scientific and Sigma Aldrich, and 1-H-pyrazole-4-carboxylic acid ($H_2PyC$) from Ambeed. All chemicals were used without further purifications. Powder X-ray diffraction (PXRD) data were collected on a Bruker AXSD8AdvanceA25 Powder X-ray diffractometer (40 kV, 40 mA) using Cu Ka (l = 1.5406 Å) radiation. Infrared (IR) spectra were recorded on a Nicolet Impact 410 FTIR spectrometer. The microscopy data were captured on a KEYENCE Fluorescence Microscope BZ-X810. Scanning electron microscopy (SEM) images were performed on a Hitachi SU 8000. UV–vis absorption was measured on a Cary 300 Bio UV–visible spectrophotometer.

### The synthesis and activation of MOF-919

$AlCl_3 \cdot 6H_2O$ (43.1 mg), Cu $(NO_3)_2 \cdot 3H_2O$ (207.1 mg), $H_2PyC$ (39.0 mg), and trifluoroacetic acid (50 µL) were totally dissolved in 10 mL of DMF and sonicated in a 20 mL Pyrex vial. The mixture was heated at 100 °C for 10 h, and cool down to room temperature for obtaining blue precipitation. The prepared MOF-919 were placed in DMF for 3 days, while the solvent exchange was conducted five times with pure DMF. This was followed by another solvent exchange using ethanol for a period of three days, 3 times per day. Finally, the solvent exchanged MOF-919 were dried under vacuum at ambient temperature for 2 h and then at 150 °C for 12 h to accomplish activated MOF-919. The obtained characterizations data were all consistent with those reported in the literature[34].

### Preparation of GFP@MOF-919

GFP-loaded MOF-919 were prepared in different concentration of GFP (18.6, 9.6, 6.2, and 3.8 mg/mL, **C1**–**C4**), following the same procedures via saturating well-dispersed MOF-919 (50 mg) in protein solution of 20 mM Tris-buffer, pH 7.5, 50 mM NaCl. Each of the samples were placed on a shaker with 300 rpm at room temperature for 12 h. Next, these samples were centrifuged at $11,000 \times g$ for 10 min and washed three times with tris buffer and D.I. water, respectively. The supernatants were measured to ensure no detectable free GFP in solution. All GFP@MOF-919 samples were vacuum-dried to remove moisture for further characterizations. h-GFP (~10 mg/mL, 200 µL) was utilized to track protein loading process in MOF-919 (~5 mg), and adsorption profile was characterized by UV–vis absorption using a Cary 300 Bio UV–visible spectrophotometer while measuring the concentration changes of free h-GFP.

### h-/d-GFP expression

Deuterated (d-GFP) and hydrogenated GFP (h-GFP) were obtained from Bio-deuteration Laboratory at the Oak Ridge National Laboratory and prepared as reported in the literature[37]. h-GFP and d-GFP were overexpressed in Escherichia coli BL21 (DE3) transformed with pET28a_AvGFP, which produced a mutant identical to the GFPMut3 variant with the exceptions of F64L, G65T, A72S, F99S, M153T, and V163A. Culture conditions and purification procedures are reported in previous work[38]. For fully deuterated GFP, 0.5% (w/v) $D_8$-glycerol was employed as the carbon source. After purification, the proteins were exchanged in $D_2O$ and lyophilized. Lyophilized fully deuterated and hydrogenated proteins were utilized as dry samples.

### SANS measurement

SANS experiments were carried out using the 30-meter-long Small-Angle Neutron Scattering instrument on neutron guide CHRNS 30 m SANS beamline at National Institute of Standards and Technology Center for Neutron Research and the CG2(GP-SANS) and the CG3(Bio-SANS) beamlines at ORNL[31,48]. Multiple configurations with different wavelengths and sample-to-detector distances were selected to cover a q-range of $0.0015 < q < 0.5$ A$^{-1}$. Samples were sandwiched by two quartz windows with 1 mm neutron path length. All the measurements were conducted at ambient conditions. The scattering intensities were corrected for empty cell scattering, sample transmission, thickness,

detector sensitivity, and instrument noise before placed on the absolute scale using a secondary standard. Finally, the 1D curves were obtained via azimuthal average from the 2D scattering data[49].

### Measurement of h-GFP solution and d-GFP solution

Approximately 5 mg/mL of h-GFP and d-GFP solutions were prepared in a 95% $D_2O$ and 50% $H_2O$ Tris-buffer, respectively, and SANS measurements of both free protein solutions were conducted in 2-mm-pathlength quartz cells (banjo cells) under same conditions. The software used for protein structural analysis, including BioXTAS RAW for Rg, P(r), GNOM, DAMMIF, etc.[50–52].

### Contrast variation of MOF-919

The aim of contrast variation experiments is to match the scattering length density of MOF-919 and it was achieved by adjusting the ratio of protonated and deuterated water. The contrast matching point of MOF-919 were determined by experimental and theoretical methods. The MOFs were immersed into mixture of $D_2O$ and $H_2O$ at different ratio of $D_2O$ and tested by SANS. 50%/50% of $D_2O/H_2O$ were finally determined for MOF-919. The correlation peaks disappear after the entering of water into the pores. The incoherent scattering background increases because of the $H_2O$.

### SANS measurements of MOF-919 and d-GFP@MOF-919

Dry and wet solid samples were characterized by following the same procedures on quartz plates in spacers with 0.43 mm path. The same amount of ~40 mg dry samples were loaded into the spacer for SANS measurement to ensure the similar scattering intensity at high and low q, from background matrix MOF-919. The wet d-GFP@MOF-919 were prepared by completely saturating these dry solid samples after the first measurement, with 50%/50% $D_2O$ and $H_2O$. The spacers then were sealed to ensure the full hydration of all d-GFP@MOF-919 during the SANS measurement.

## Data availability

The authors declare that all the data supporting the findings of this study are available within the article and Supplementary Information Files, and/or from the corresponding authors on request. Source data are provided with this paper.

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

## Acknowledgements

The authors acknowledge the financial support from the US National Science Foundation (DMR-1352065) and the Robert A. Welch Foundation (B-0027). The SANS experiments were performed using beamlines of Bio-SANS (CG3) and GP-SANS (CG2) at the High Flux Isotope Reactor in ORNL and NG 7-SANS instrument in NIST. Neutron scattering experiments on Bio-SANS were supported by the Center for Structural Molecular Biology funded by DOE Biological and Environmental Research (project ERKP291). A portion of this research used resources at the High Flux Isotope Reactor, a DOE Office of Science User Facility operated by the Oak Ridge National Laboratory. Access to VSANS was provided by the Center for High Resolution Neutron Scattering, a partnership between the National Institute of Standards and Technology and the National Science Foundation under Agreement No. DMR-2010792. Partial support from the Researchers Supporting Program project no (RSP2023R55) at King Saud University, Riyadh, Saudi Arabia is also acknowledged (A.M.A.). We appreciate Prof. Hexiang Deng and Gaoli Hu for generously providing the MOF-919 material. Disclaimer: The identification of any commercial product or trade name does not imply endorsement or recommendation by the National Institute of Standards and Technology.

## Author contributions

S.M. and L.H. conceived and designed the research. X.W. carried out the experimental works. H.O.N. and Q.Z. expressed and provided the h-GFP and d-GFP samples. Protein reconstructions were performed by S.Q. and J.S., and Y.M. assisted the SANS experiments in NIST. C.C. helped with the characterization of gas adsorption. A.M.A. and A.N. assisted to revise the manuscript. All authors participated in drafting the paper, and gave approval to the final version of the manuscript.

## Competing interests

The authors declare no competing interests.
