## [Peer Review File · Nature Communications]

REVIEWERS' COMMENTS

Reviewer #1 (Remarks to the Author):

The manuscript of X. Wang et al. presents a structural study of GFP protein confined in mesoporous metal-organic framework (MOF). The authors intend to demonstrate that structural information can be obtained on the conformation and the oligomerization state of GFP from small-angle neutron scattering (SANS) by contrast matching MOF with isotopic H/D substitution of aqueous buffers and deuteration of the GFP protein.

The study is based on the scattering sets of the GFP protein at four different concentrations in presence of "contrast-matched" MOF compared to aqueous solutions of free GFP. The analysis of the SANS data is achieved mainly by Monte Carlo construction of models in real space through the assembly of dummy beads, often called *ab initio* reconstruction.

The presentation of the data analysis is too elliptical. Mainly, there is no real conclusion about the interaction between GFP and MOF. The authors should thoroughly review the manuscript before resubmitting it for publication. The current version is not acceptable.

The authors must consider the following points.

Major points:

- In general, the manuscript is not well written and does not contain enough numeric values (R_g , etc.). Some tables present in the SI could be present in the main text. The supplementary work (Figs 1, 4, 5, 7-12, 16-17, 20 and Tables 3-5) not cited in the text should be deleted.

- Equation 1: the word "Lorentzian" must be avoided if m (Eq. 1) is not equal to 2, because of the physical meaning associated to its Fourier transform (FT).

What is the meaning of the large value of m ? I do not see the physical meaning of ξ either. Both the mathematical equation and the fitted value of ξ , which is larger than the periodicity of the MOF, should be explained.

Actually, I think that the two first terms of Eq. 1 are empirical representations of the large scattering of the matrix. Also, the equation does not play any role in the discussion!

- MOF is poorly contrast matched, likely due to the large amount of dry pores. A numerical evaluation of the ratio filled/empty pores is possible from the two contrasts contributing to the Porod scattering.

- Since it is not just a matter of subtracting a constant but rather the contribution of the large scattering due to the interfaces in MOF, the procedure is not quite correct. Indeed, the subtraction implies the absence of interactions between protein and MOF, in contradiction with the conclusions. Therefore, the limitations of this approximation must be mentioned.

In Fig. 2, "decoupling" must be replaced by "subtraction of the MOF scattering".

- *Ab initio* reconstructions: how is the modelization achieved, in particular the formation of oligomers? It is not clear if the interactions are only between proteins or also with MOF. What is the assumed interaction between GFP and MOF?

Are the proteins simply inserted in the MOF framework (in blue?) in Figs 19-22 & Fig. 4b.

- In order to check the effect of MOF, the correct control of free GFP should be performed at the same concentration as in the MOF and not only at the low concentration of 5 mg/mL. What is the scattering of free GFP at concentrations 9.6 and 18.6 mg/mL?

- Free GFP protein is a beta barrel as correctly described in the last paragraph of the Introduction. However, later, the authors interpret the $P(r)$ data as due to a sphere. This contradiction must be clarified, especially the values of R_g . How are R_g calculated? Fig. 3a must be compared with the theoretical curve that can be obtained from the PDB file of GFP.

- Since the contrast is squared, the coherent scattering of h-GFP/D₂O and d-GFP/H₂O should be identical (Babinet's principle). The eventual effect of labile atoms at the interface cannot explain

the differences observed in Fig. 3. I think that the difference between the two curves is more likely due to the subtraction of the background. A fitting of $I(q)$ would be more significant than its FT.

- Fig. 18 in the SI: the invariants can be calculated from the area under the curves in each original Kratky plot (IQ^2 as function of Q). It gives a reliable information, namely allowing the evaluation of the volume, thus the degree of oligomerization of the proteins.

Minor points:

- "superstructure": this term, meaning a precise organization, is excessive in the context.

- The curves in Fig. 2 should be fitted with a form factor.

- Fig. 4 (main text) and Fig. S6 (SI): the differences between these two figures must be clarified.

- What do the slow and fast modes mean (Fig. 15 SI) in DAMMIF? Do the authors test also MASSHA from ATSAS?

- Page 12: it is not clear what the authors mean by "distance" in the sentence "all of the simulated arrangements have a peak representing the GFP monomer around 35-40 Å, while the most probable distance of the free d-GFP in solution is peaked at around 30 Å".

- Discussion: what is the difference (advantages/drawbacks) between MOF and other mesoporous media for the confinement of proteins?

- Discussion: GFP is presented as a « model » protein: how can the results found here be generalized to other proteins or to other protein families?

- The choice of references has to be precised or checked, especially:

- References 18-23 are related to the structural study of proteins but especially in MOF.

- References 24-30 seem to not be related to MOF, except reference 26.

- References 31-32 do not concern nanopores.

- Reference 40 is missing: it is part of ref. 39.

Reviewer #2 (Remarks to the Author):

The authors have used SANS to directly visualize GFP encapsulated in the nanospace of MOFs. Contrast matching with deuterated GFP was used to distinguish the protein component from hydrogenated MOF. The data are original and the findings are certainly impressive, especially that this experiment requires a high preparative effort. However, a number of shortcomings should be considered before the manuscript can be accepted for publication:

1. The authors report in detail that there are three different types of pores in the MOF used, while the data analysis requires monodispersity. This contradiction should be explained in more detail.

2. It is mentioned that the plateau observed in the SANS data is a signature of monodispersity. However, this is a necessary, but no sufficient condition to conclude monodispersity. Is there any other evidence?

3. The term "superstructure" should be better defined and put into relation with the known crystal structure of GFP.

4. The data can be compared to the GFP crystal structure, e.g. using cryson.

5. The quality of figures and captions is not too high, axis labels are difficult to read, symbols are hard to distinguish. Most importantly, the data and structures shown are not sufficiently described and related to the text. In Fig. 4 color coding seems to be wrong.

Reviewer #3 (Remarks to the Author):

The authors successfully presented the spatial arrangement of GFP inside MOF-919 using in situ small-angle neutron scattering and pair distance distribution function analysis. Verifying the protein conformation inside the mesoporous framework is crucial to show that the structural conformation of immobilized proteins remain intact, to ensure their catalytic and biological activities are well-maintained. This paper opens up the potential for the use of SANS to characterize not only proteins but even other biomolecules entrapped in porous frameworks. There are a few questions need to be addressed before the manuscript can be accepted for publication:

1. Based on the adsorption profile in Fig S2, what is the mechanism of adsorption for the GFP into MOF-919?
2. Does the initial loading concentration correlate with the actual GFP loaded inside the MOF?
3. As protein conformation may be visualized inside the MOF pore, could the data for GFP loading percentage also be obtained using SANS?
4. Kindly elaborate on how the pair distance distribution function analyses were carried out.

Point-by-Point Response to Reviewers' Comments

We greatly appreciate the constructive comments/suggestions from all reviewers, and we have revised the manuscript accordingly to fulfill their requests as detailed in the responses below. The corresponding changes have been highlighted in yellow in the main text.

Reviewer #1:

The manuscript of X. Wang et al. presents a structural study of GFP protein confined in mesoporous metal-organic framework (MOF). The authors intend to demonstrate that structural information can be obtained on the conformation and the oligomerization state of GFP from small-angle neutron scattering (SANS) by contrast matching MOF with isotopic H/D substitution of aqueous buffers and deuteration of the GFP protein.

The study is based on the scattering sets of the GFP protein at four different concentrations in presence of “contrast-matched” MOF compared to aqueous solutions of free GFP. The analysis of the SANS data is achieved mainly by Monte Carlo construction of models in real space through the assembly of dummy beads, often called ab initio reconstruction.

The presentation of the data analysis is too elliptical. Mainly, there is no real conclusion about the interaction between GFP and MOF. The authors should thoroughly review the manuscript before resubmitting it for publication. The current version is not acceptable.

Response: We are grateful to the reviewer for taking time to evaluate our work. We have thoroughly revised our manuscript to clearly address the reviewer's comments as detailed below.

However, we respectfully disagree with the reviewer's comments “The presentation of the data analysis is too elliptical. Mainly, there is no real conclusion about the interaction between GFP and MOF”. It seems that the reviewer might misunderstand the theme of our work, which aims to find protein arrangement inside MOF using deuterated protein (i.e. GFP) to increase scattering contrast between loaded protein and matrix MOF. Nonetheless, the interactions of protein-protein or protein-MOF are elusive in this work, and it is very tough to use the technique of SANS in the system of protein-MOF. The protein arrangement was concluded by reconstructing proteins inside MOFs after signaling out the scattering from d-GFP. Considering the pair distance among protein assembly inside confined MOF pores, we assumed that they are likely to form interactions among proteins or with the MOF wall which caused the aggregation of proteins. But the clarification of detailed interactions and mechanisms is out of the scope of this work.

Comment 1: In general, the manuscript is not well written and does not contain enough numeric values (R_g , etc.). Some tables present in the SI could be present in the main text. The supplementary work (Figs 1, 4, 5, 7-12, 16-17, 20 and Tables 3-5) not cited in the text should be deleted.

Response: We agree with the reviewer and thank the reviewer for pointing this out. We have moved table S4 to the main text and discussed the Figs of supplementary information in the main text as well.

Comment 2: Equation 1: the word “Lorentzian” must be avoided if m (Eq. 1) is not equal to 2, because of the physical meaning associated to its Fourier transform (FT).

Response: We agree with the reviewer and thanks for pointing this out. That term becomes the well-known Lorentzian function only if $m=2$. We have changed “Lorentzian” to “Exponent m ” in Supplementary Table 1, Table 2, and Table 3.

Comment 3: What is the meaning of the large value of m ? I do not see the physical meaning of ξ either. Both the mathematical equation and the fitted value of ξ , which is larger than the periodicity of the MOF, should be explained.

Response: We appreciate the constructive comments from the reviewer. The correlation length model is an empirical function form for the analysis of SAS data for randomly distributed two-phase systems. (please see the following articles). Our original fitting of this disordered structure at a larger level using the Debye-Anderson-Brumberger (DAB) function was not successful due to the hierarchical structure of the material. Therefore, we turned to this empirical model. (Supporting refs: https://www.sasview.org/docs/user/models/correlation_length.html and B Hammouda, D L Ho and S R Kline, *Macromolecules*, 37, 6932-6937(2004)).

In the DAB model, ξ is a measure of the average spacing between two regions, which is not the repeating distance of the pore arrangement in the MOF. The parameter of ξ is very similar in this empirical model. It reflects the inhomogeneity of the materials at larger scale structure. In this work, our purpose using this model is to decouple the scattering contributions from different length scales. Eventually, the scattering contribution from the confined proteins is singled out for the subsequent 3D reconstruction. We should avoid the overstating of the meanings of these parameters.

Comment 4: Actually, I think that the two first terms of Eq. 1 are empirical representations of the large scattering of the matrix. Also, the equation does not play any role in the discussion!

Response: The first term describes the Porod scattering from the surface of the MOF granule particles, which is not an empirical function. The second term is an empirical function. Our purpose using these terms was to decouple the scattering contributions on different length scales. Eventually, the scattering contribution from the confined protein molecules is singled out for further analysis. We should avoid the overstating of the meanings of these parameters. Therefore, we did not discuss the fitting results using Eq. 1. Instead, we performed 3D reconstruction upon the scattering contribution of the protein upon is singled out.

Comment 5: MOF is poorly contrast matched, likely due to the large amount of dry pores. A numerical evaluation of the ratio filled/empty pores is possible from the two contrasts contributing to the Porod scattering.

Response: The MOF exhibits a hierarchical structure with multiple length scales and contains disordered large size macropores and large-scale inhomogeneity besides the ordered mesopores. It is not an ideal two-phase system. The contrast matching point is different at different length scales. Therefore, it is impossible to have a perfect contrast matching. The residual scattering arises largely from the inaccessible macropores. One can evaluate the ratio of filled/empty pores according to the Porod Invariant if the amount of material exposed to the neutron beam doesn't change before and after the penetration of protein and water. However, in the experiment, it is very challenging to keep that amount invariant at dry and wet states.

Comment 6: Since it is not just a matter of subtracting a constant but rather the contribution of the large scattering due to the interfaces in MOF, the procedure is not quite correct. Indeed, the subtraction implies the absence of interactions between protein and MOF, in contradiction with the conclusions. Therefore, the limitations of this approximation must be mentioned.

Response: The high q scattering data is from two contributions: one is incoherent scattering because of hydrogens in the MOF, protein and the contrast solvent, which is q independent; the other contribution is from the interface between two phases of the material, which is q dependent. In SANS experiments, the incoherent scattering constant is obtained through data fitting, which has been a routine method in the SANS community. (Please see Rubinson K. A et al. J. Appl. Cryst. (2008). 41, 456–465.). As can be seen from Figure 1d, the scattering curve becomes flat at high q, which means the subtraction of the incoherent scattering in our work is trustworthy. The subtraction of this incoherent scattering background doesn't mean the interactions between the protein and the MOF matrix are ignored; instead, the removal of incoherent scattering highlights the scattering from the interface.

Comment 7: In Fig. 2, “decoupling” must be replaced by “subtraction of the MOF scattering”.

Response: We appreciate the suggestion from the reviewer. We have made the revision per the suggestion.

Comment 8: Ab initio reconstructions: how is the modelization achieved, in particular the formation of oligomers? It is not clear if the interactions are only between proteins or also with MOF. What is the assumed interaction between GFP and MOF?

Response: The protein arrangement is concluded by the reconstruction of protein inside MOFs. Once the ab initio results of the envelope are obtained, the PDB structure of GFP is filled into the volume result with the constraint of MOF cage structure and layout to further mimic the shape and dimension of d-GFP in MOF-919. Then the simulated GFP ‘multimer’ configuration is used to generate calculated scattering with CRYSON. Whether the protein-protein or protein-MOF interactions result in this protein assembly is still elusive in this work, and it is difficult to clarify by the technique of SANS in the system of protein-MOF. MOF scattering is suppressed compared to the GFP and there isn't a major contribution from the MOF in the q-range that that GFP dominates. Considering the pair distance among protein assembly in MOF, we assumed there are existing non-covalent interactions (e.g. hydrogen bonding and π - π interaction) among proteins. The clarification of detailed interactions and mechanism is out of the scope of this work. However, this study will undoubtedly assist our following works to investigate the existing interactions.

Comment 9: Are the proteins simply inserted in the MOF framework (in blue?) in Figs 19-22 & Fig. 4b.

Response: Considering the dimension and shape of GFP and MOF pores, we simulated to fit GFPs with proper arrangement inside MOF cavities. e. g. GFP is a cylinder with a length of 4.2 nm and a diameter of 2.4 nm and MOF-919 contains two meso-cages of 4.9 nm and 6.0 nm, in which two types of mesocages are fused through the pentagonal aperture of 2.0 nm, and 2.4 nm hexagonal openings for two larger nanocages, respectively. The best fitting arrangement of GFP is in a large liu cage, and this is also mentioned in the main text.

Comment 10: In order to check the effect of MOF, the correct control of free GFP should be performed at the same concentration as in the MOF and not only at the low concentration of 5 mg/mL. What is the scattering of free GFP at concentrations 9.6 and 18.6 mg/mL?

Response: The concentrations of 9.6 and 18.6 mg/mL were the initial loading concentrations before being immersed with a specific amount of well-dispersed MOF-919 (50 mg), which were not the loaded protein concentration in the MOF. The stock GFP solution (~20 mg/mL) were serially diluted into different concentration of C1-C4 (18.6, 9.6, 6.2 and 3.8 mg/mL). The higher GFP concentration will lead to higher loading efficiency and amount, which is supported by the increasing scattering from loaded SANS GFP. Thus, our study is to investigate the differences in protein arrangement with varied loading amounts.

Fig. 6. Small angle neutron scattering (SANS) curves of three different concentrations of deuterated m-GFP in hydrogenous buffer, at 1 mg/mL (green – diamond), 5 mg/mL (red – square) and 10 mg/mL (black – ellipse) concentrations. The spectra is shown in Log I(q) versus Log q. Data points are plotted where the error bars are less than the value of the data point. The protein is in 20 mM Phosphate, 150 mM NaCl buffer in ddH₂O. The inset shows a Guinier plots of m-eGFP for the SANS curves with the calculated radius of gyration (Rg) and intensity at zero angle (I(0)).

Myatt D. P. et al performed free GFP in water and their results showed that the dimer increased as a percentage of the total concomitant to increasing protein concentration, which can be seen from the increasing Rg (from 17.17~20.8 Å) they obtained (please see figure above, also see Myatt D. P. et al Biomedical Spectroscopy and Imaging 6 (2017) 123–134). The values of Rg increased concomitantly with concentration. In our data, the Rg of the structure with loading concentration C2 (9.6mg/mL) is already 30.69Å, which is the strong evidence that the confinement leads to the assembly of the proteins.

Comment 11: Free GFP protein is a beta barrel as correctly described in the last paragraph of the Introduction. However, later, the authors interpret the P(r) data as due to a sphere. This contradiction must be clarified, especially the values of Rg. How are Rg calculated? Fig. 3a must be compared with the theoretical curve that can be obtained from the PDB file of GFP.

Response: We described the GFP protein as spherical in the view of small-angle scattering is not in contradiction with its beta barrel structure. Typically, at resolution of SAS level (10-30 Å), globular or spherical particle is used to describe the factor that the protein is a compact and folded particle that is not extendedly elongated like a long rod or thin-sheet. The Rg is calculated by a linear fitting from ln(I) vs. q² through $I(q) \approx I(0)\exp(-q^2 R_g^2/3)$ in software BioXTAS RAW.

Comment 12: Since the contrast is squared, the coherent scattering of h-GFP/D₂O and d-GFP/H₂O should be identical (Babinet's principle). The eventual effect of labile atoms at the interface cannot

explain the differences observed in Fig. 3. I think that the difference between the two curves is more likely due to the subtraction of the background. A fitting of $I(q)$ would be more significant than its FT.

Response: According to the neutron scattering length density (NSLD) of the proteins and solvent (D_2O or H_2O) used in the study, the contrasts, which is the difference in the NSLD are not exactly like Babinet's principle. Here, h-GFP NSLD $\sim 2.348 \times 10^{-6} \text{ \AA}^{-2}$, D_2O $6.335 \times 10^{-6} \text{ \AA}^{-2}$, d-GFP $\sim 6.9 \times 10^{-6} \text{ \AA}^{-2}$, H_2O : $\sim -0.561 \times 10^{-6} \text{ \AA}^{-2}$, so the contrasts are not the same. However, the differences in structure and R_g are known effect on the contrast and hydration shell effect as we mentioned in ref. 37 (Perticaroli, S. et al. *J Am Chem Soc* 139, 1098-1105 (2017)). Also, the isotope effect may lead to different conformation of proteins. Efimova Y. M. et al. observed that protein lysozyme adopted a more compact conformation in D_2O than in H_2O , which seems consistent with our results. (Efimova Y. M. et al. *Biopolymers* 85, 264-273 (2007))

Comment 13: Fig. 18 in the SI: the invariants can be calculated from the area under the curves in each original Kratky plot (IQ^2 as function of Q). It gives a reliable information, namely allowing the evaluation of the volume, thus the degree of oligomerization of the proteins.

Response: We already calculated the Porod volume and listed them in the Supplementary Table 5 in our initial submission.

Comment 14: "superstructure": this term, meaning a precise organization, is excessive in the context.

Response: We have used protein "assembly" to replace "superstructure".

Comment 15: The curves in Fig. 2 should be fitted with a form factor.

Response: Only the free GFP data can be fitted with an elliptical cylinder form factor, which is shown below. Evidently, this form factor cannot fit confined proteins. Therefore, we use the reconstruction technique for the data analysis.

Comment 16: Fig. 4 (main text) and Fig. S6 (SI): the differences between these two figures must be clarified.

Response: We thank the reviewer for pointing this out. Fig.4 is the scattering profile of wet MOF-919 at the contrast matching point, and Fig. S6 is dry materials where solvent has been removed.

Comment 17: What do the slow and fast modes mean (Fig. 15 SI) in DAMMIF? Do the authors test also MASSHA from ATSAS?

Response: According to DAMMIN/DAMMIF manual, 'Configuration of the annealing procedure, one of Fast (bigger beads, cooling down quickly), Slow (smaller beads, cooling down slowly),' it was discussed in D. I. Svergun (1999). Restoring low resolution structure of biological

macromolecules from solution scattering using simulated annealing. Biophys J. 2879-2886 (1999). We have added the reference in the SI. MASSHA appears to be obsolete as a rigid body modeling tool in ATSAS as the group stopped releasing it in newer version. Our case of the GFP assembly or multimer is also a different case that typical rigid body modeling can handle as they are now in the form of typical noncovalently bond protein oligomer but assembly constraint by MOF structure and cage locations as shown in Fig.4 and SI Fig.

Comment 18: Page 12: it is not clear what the authors mean by "distance" in the sentence "all of the simulated arrangements have a peak representing the GFP monomer around 35-40 Å, while the most probable distance of the free d-GFP in solution is peaked at around 30 Å".

Response: We agree with the reviewer and thank the reviewer for pointing this out. The distance means 'the pair distance'. We have corrected that.

Comment 19: Discussion: what is the difference (advantages/drawbacks) between MOF and other mesoporous media for the confinement of proteins?

Response: The emergence of MOF greatly extended the path of immobilizing enzymes because of the unique aspects of MOFs like crystallinity nature, high porosity, open active sites, versatile synthetic conditions, and tunable structure. The periodic and consistent microenvironment from MOFs can provide more stable interactions with the immobilized protein. Among those traditional porous materials composed of either organic or inorganic components, intrinsic limitations still exist, specifically, inorganic porous materials lack structural flexibility, and organic porous materials tend to be structurally amorphous. The combination of biomolecules and MOFs can integrate both principal properties into one synergistic system without compromise. Also, the high crystallinity of MOF is easy to modelized and characterize by SANS.

Comment 20: Discussion: GFP is presented as a « model » protein: how can the results found here be generalized to other proteins or to other protein families?

Response: We have added the additional discussions. The aim of this work is to find protein arrangement inside MOF via using deuterated protein to increase scattering contrast among loaded protein and matrix materials. Actually, considering the NSLD of other proteins, they have similar value and contrast matching point of approximately 40% D₂O, and is close to MOF-919 (~50% D₂O). The availability of deuterated GFP allowed this study to be done and the new knowledge about its distribution in the MOF can be applied to different types of proteins.

Comment 21: The choice of references has to be precised or checked, especially:

- References 18-23 are related to the structural study of proteins but especially in MOF.
- References 24-30 seem to not be related to MOF, except reference 26.
- References 31-32 do not concern nanopores.
- Reference 40 is missing: it is part of ref. 39.

Response: We thank the reviewer for pointing this out. These refs have been corrected.

Reviewer #2:

The authors have used SANS to directly visualize GFP encapsulated in the nanospace of MOFs. Contrast matching with deuterated GFP was used to distinguish the protein component from

hydrogenated MOF. The data are original and the findings are certainly impressive, especially that this experiment requires a high preparative effort. However, a number of shortcomings should be considered before the manuscript can be accepted for publication:

Response: We are grateful to the reviewer for taking time to evaluate our work and appreciate the high comments and support from the reviewer.

Comment 1: The authors report in detail that there are three different types of pores in the MOF used, while the data analysis requires monodispersity. This contradiction should be explained in more detail.

Response: Like many other MOF materials, the studied MOF exhibits hierarchical structure, which can be seen from the scattering curves of dry sample. However, we used a model with multiple terms to account for the structural features at different length scales. Eventually, we decomposed the scattering curve and obtained the scattering only from the confined protein for further data analysis. Therefore, the data analysis doesn't require monodispersed pores.

Comment 2: It is mentioned that the plateau observed in the SANS data is a signature of monodispersity. However, this is a necessary, but no sufficient condition to conclude monodispersity. Is there any other evidence?

Response: The plateau shown in the scattering curves of Figure (4b) is already after the scattering decoupling. How we performed the decoupling using multiple level structures can be found from the response to Comment 1.

Comment 3: The term "superstructure" should be better defined and put into relation with the known crystal structure of GFP.

Response: We thank the reviewer for pointing this out. We have used protein "assembly" to replace "superstructure", and it has been defined in the main text.

Comment 4: The data can be compared to the GFP crystal structure, e.g. using cryson.

Response: We appreciate for this suggestion. The CRYSON has been added in the main text.

Comment 5: The quality of figures and captions is not too high, axis labels are difficult to read, symbols are hard to distinguish. Most importantly, the data and structures shown are not sufficiently described and related to the text. In Fig. 4 color coding seems to be wrong.

Response: We thank the reviewer for pointing this out. The figures and caption are re-made, and Fig. 4 is replaced with new color coding.

Reviewer #3:

The authors successfully presented the spatial arrangement of GFP inside MOF-919 using in situ small-angle neutron scattering and pair distance distribution function analysis. Verifying the protein confirmation inside the mesoporous framework is crucial to show that the structural conformation of immobilized proteins remain intact, to ensure their catalytic and biological activities are well-maintained. This paper opens up the potential for the use of SANS to characterize not only proteins but even other biomolecules entrapped in porous frameworks. There are a few questions need to be addressed before the manuscript can be accepted for publication:

Response: We are grateful to the reviewer for taking time to evaluate our work and appreciate the high comments and support from the reviewer.

Comment 1: Based on the adsorption profile in Fig S2, what is the mechanism of adsorption for the GFP into MOF-919?

Response: For protein immobilization, the main driving force is free diffusion due to the protein concentration difference. Also, the interaction of protein-MOF and confinement effects from matrix support play a critical role in protein encapsulation and immobilized GFP can be well retained in the cavities of MOF-919. From some earlier works (J. Am. Chem. Soc. 134, 13188–13191(2012)) and Chem. Sci. 10, 4082–4088 (2019), the partial unfolding due to the flexible nature of protein can drive them into deep MOF pores when the MOF open apertures are close or slightly smaller than the protein diameter.

Comment 2: Does the initial loading concentration correlate with the actual GFP loaded inside the MOF?

Response: Based on the scattering profiles we collected, MOF-919 (C1-C4), the highest loading concentration (C1) exhibits the strongest scattering signal and largest protein assembly in MOF (mixture of trimer and tetramer). However, the lowest initial concentration(C4) forms a monomer or dimer inside MOF with low scattering (Fig. 2). It is believed that the high initial loading concentration led to higher loading amount in this work.

Comment 3: As protein conformation may be visualized inside the MOF pore, could the data for GFP loading percentage also be obtained using SANS?

Response: The scattering intensity is proportional to the sample volume exposed to the neutron beam. In our work, the powder samples were packed in demountable cells. The packing density may vary between the samples. Therefore, it is difficult to quantitatively calibrate protein loading amounts based on scattering profiles.

Comment 4: Kindly elaborate on how the pair distance distribution function analyses were carried out.

Response: The P(r) analysis was carried out in the software BioXTAS RAW. We have added references for the software. (Journal of Applied Crystallography 25, 495-503 (1992), Biophysical Journal 76, 2879–2886 (1999) and Journal of Applied Crystallography 50, 1545-1553 (2017))

Again, we thank the reviewers for the constructive comments/suggestions, which have made our manuscript substantially improved.

Sincerely,

Shengqian Ma, PhD
Professor and Welch Chair in Chemistry

REVIEWERS' COMMENTS

Reviewer #1 (Remarks to the Author):

The authors took into account many of my comments and suggestions.

There remains a misunderstanding on my comments 6 and 8. It is not correct that "MOF scattering is suppressed compared to the GFP" (rebuttal of comment 8). The total scattered intensity is not the addition of MOF and GFP scattering. Cross terms are important in a porous matrix, which cannot be treated as a sample holder. Erasing the contrast between the solvent and the MOF does not eliminate the interactions. My comment did not refer to the always-correct subtraction of the incoherent scattering at large q -values.

Consequently, the measurement is correct but the conclusion is limited to the experimental conditions and adjusted by empirical parameters without physical significance as the authors admit (answer to comment 4).

Despite this limitation, the article can be accepted for publication.

Reviewer #2 (Remarks to the Author):

The authors have revised their paper, however some questions have not been answered satisfactorily and should be addressed again more thoroughly. For a paper in NComms I would also expect to here a bit more about a possible impact of the work for a wider community.

I refer to the following original comments and answers by the authors:

Comment 2 (initial review): It is mentioned that the plateau observed in the SANS data is a signature of monodispersity. However, this is a necessary, but no sufficient condition to conclude monodispersity. Is there any other evidence?

Response: The plateau shown in the scattering curves of Figure (4b) is already after the scattering decoupling. How we performed the decoupling using multiple level structures can be found from the response to Comment 1.

This is no answer to the question, unfortunately. It is well understood that the authors had to achieve a de-coupling of the MOF- and GFP-contributions. However, the plateau in the remaining GFP data is still no confirmation of monodispersity- This is especially important as the authors show different GFP sizes at different load factors (see figure 4). It is well-known that the SANS contributions of different oligomers of the same protein can overlap and yield a plateau without monodispersity. So, if there is no independent proof, it should be stated that monodispersity is assumed to enable the data analysis as performed here.

Comment 5 (initial review): The quality of figures and captions is not too high, axis labels are difficult to read, symbols are hard to distinguish. Most importantly, the data and structures shown are not sufficiently described and related to the text. In Fig. 4 color coding seems to be wrong.

Response: We thank the reviewer for pointing this out. The figures and caption are re-made, and Fig. 4 is replaced with new color coding.

Also this point has not been addressed sufficiently. The figures still have axis labels so small that they cannot be read, see e.g. figures 1 and 2.

In figure 2, green and cyan color can hardly be distinguished, some legends have units, some do not have units.

A hump at 0.04 \AA^{-1} described in the caption/text cannot be seen. Is this a problem of small labels, because of an error or because of something else?

minor remarks:

The peaks given in Table 1 are not assigned in the figure.

Correct several typos like "dimmer", "umol" or "owning" and check the whole text for others.

Reviewer #3 (Remarks to the Author):

The authors have significantly revised the manuscript to reflex the comments from reviewers. I agreed with the responses from the authors. The manuscript in this recent form, therefore, can be accepted now.

Point-by-Point Response to Reviewers' Comments

We are greatly thankful to the approval for publication from reviewer 1 and 3 and more constructive comments/suggestions from Reviewer 2. We have redrawn all the figures as Reviewer 2 requested and substantially revised the manuscript to fully address his/her concerns as detailed in the following responses to **Reviewer 2's** comments.

Reviewer #2:

Comment 1: Comment 2 (initial review): It is mentioned that the plateau observed in the SANS data is a signature of monodispersity. However, this is a necessary, but no sufficient condition to conclude monodispersity. Is there any other evidence?

Response: The plateau shown in the scattering curves of Figure (4b) is already after the scattering decoupling. How we performed the decoupling using multiple level structures can be found from the response to Comment 1.

This is no answer to the question, unfortunately. It is well understood that the authors had to achieve a de-coupling of the MOF- and GFP-contributions. However, the plateau in the remaining GFP data is still no confirmation of monodispersity- This is especially important as the authors show different GFP sizes at different load factors (see figure 4). It is well-known that the SANS contributions of different oligomers of the same protein can overlap and yield a plateau without monodispersity. So, if there is no independent proof, it should be stated that monodispersity is assumed to enable the data analysis as performed here.

Response: We agree with the reviewer. The Guinier plateau is not sufficient condition to indicate the monodispersed proteins confined in the pores. It could be monodispersed or multimers overlapped with smaller oligomers like the reviewer mentioned. In the later analysis, we concluded that the size of protein assembly could be mix of "trimers" and "dimers". Thus, we have deleted this statement regarding monodispersity in the revised version and changed it to "The presence of the Guinier plateau in the scattering profiles indicates that the "assembly" formed by the confined proteins is not hierarchical and free of network structure".

Comment 2: Comment 5 (initial review): The quality of figures and captions is not too high, axis labels are difficult to read, symbols are hard to distinguish. Most importantly, the data and structures shown are not sufficiently described and related to the text. In Fig. 4 color coding seems to be wrong. Response: We thank the reviewer for pointing this out. The figures and caption are re-made, and Fig. 4 is replaced with new color coding.

Also this point has not been addressed sufficiently. The figures still have axis labels so small that they cannot be read, see e.g. figures 1 and 2.

Response: We have redrawn all the figures (1-4) in main text with larger labels and higher resolution.

Comment 3: In figure 2, green and cyan color can hardly be distinguished, some legends have units, some do not have units.

Response: We thank the reviewer for pointing this out. This issue has been resolved. The cyan color has been replaced with blue and the figure has been replaced.

Comment 4: A hump at 0.04 Å⁻¹ described in the caption/text cannot be seen. Is this a problem of small labels, because of an error or because of something else?

Response: We have added the orange arrow to indicate the hump in Figure 2a. The hump peak is within a wide range (0.03-0.05 Å⁻¹) and we use ~0.04 Å⁻¹ to represent the approximate location. It is easier to distinguish from red line with the black one.

minor remarks:

Comment 5: The peaks given in Table 1 are not assigned in the figure.

Response: We thanks the reviewer for pointing this out. These peaks (a-c) of C1-C4 are labeled in each P(r) curve of Supplementary Figure 18-21, respectively, and it is also mentioned in main text.

Comment 6: Correct several typos like "dimmer", "umol" or "owning" and check the whole text for others.

Response: We thanks the reviewer for pointing this out. The manuscript has been thoroughly polished, and the typos were corrected.

Again, we thank Reviewer 2 for the constructive comments/suggestions, which have made our manuscript further improved.

Sincerely,

Shengqian Ma, PhD
Professor and Welch Chair in Chemistry

REVIEWERS' COMMENTS

Reviewer #2 (Remarks to the Author):

The authors have revised the paper so that it can now be recommended for publication.